# Investigating T Cell Immunity in Cancer: Achievements and Prospects

**DOI:** 10.3390/ijms22062907

**Published:** 2021-03-12

**Authors:** Zhen Zeng, Hui Yi Chew, Jazmina G. Cruz, Graham R. Leggatt, James W. Wells

**Affiliations:** Diamantina Institute, Faculty of Medicine, The University of Queensland, Brisbane, QLD 4102, Australia; jenny.zeng@uq.edu.au (Z.Z.); h.chew@uq.edu.au (H.Y.C.); j.gonzalezcruz@uq.edu.au (J.G.C.); g.leggatt@uq.edu.au (G.R.L.)

**Keywords:** CD8 T cells, cytotoxic T lymphocytes, cancer, immunotherapy, CAR, tumour antigen, checkpoint inhibitor

## Abstract

T cells play a key role in tumour surveillance, both identifying and eliminating transformed cells. However, as tumours become established they form their own suppressive microenvironments capable of shutting down T cell function, and allowing tumours to persist and grow. To further understand the tumour microenvironment, including the interplay between different immune cells and their role in anti-tumour immune responses, a number of studies from mouse models to clinical trials have been performed. In this review, we examine mechanisms utilized by tumour cells to reduce their visibility to CD8^+^ Cytotoxic T lymphocytes (CTL), as well as therapeutic strategies trialled to overcome these tumour-evasion mechanisms. Next, we summarize recent advances in approaches to enhance CAR T cell activity and persistence over the past 10 years, including bispecific CAR T cell design and early evidence of efficacy. Lastly, we examine mechanisms of T cell infiltration and tumour regression, and discuss the strengths and weaknesses of different strategies to investigate T cell function in murine tumour models.

## 1. Tumour-Specific T Cell Responses

As a part of the body’s immunosurveillance mechanism, all nucleated cells display major histocompatibility (MHC) class I molecules, enabling the routine presentation of cellular peptide fragments on their plasma membrane. Internal proteins are broken up by proteasomes to form peptides that are then transported by transporter associated with antigen processing (TAP) to MHC I molecules in the lumen of the endoplasmic reticulum. The peptide–MHC I complexes generated via these antigen processing and presentation pathways can be recognized by T cell receptors (TCRs) on Cytotoxic T lymphocytes (CTLs). Tumour-specific peptides are now known to exist, and to be capable of triggering spontaneous T cell responses in patients [1,2,3,4,5]. Upon detection of tumour-specific, ‘foreign’ antigens, CTLs can become activated leading to the destruction of tumour cells presenting target antigens. However, tumour cells may evade this pathway of immune-mediated destruction through multiple mechanisms.

The microenvironment within solid tumours is often hostile to tumour-specific T cells, negatively affecting their function and survival. Metabolic alterations to tumour and stromal cells result in the overexpression of glucose transporters and key glycolytic enzymes, altered growth factor signalling, hypoxic conditions, increases in acidity, and abnormal angiogenesis. Furthermore, an abundance of suppressive immune cells, including regulatory T cells (Tregs), myeloid-derived suppressor cells, and tumour-associated macrophages, infiltrate into the tumour and deliver a host of negative signals to tumour-specific T cells and antigen-presenting cells including dendritic cells. As these features of the tumour microenvironment have been reviewed extensively elsewhere (see Refs [6,7,8,9,10,11]), we will focus here on an examination of alterations to antigen processing and presentation pathways [12,13,14], factors influencing T cell infiltration, and strategies designed to improve tumour-targeting by T cells in the sections below.

### 1.1. Antigen Processing and Presentation in Tumours

Interference with the expression level of single or multiple components of antigen processing and presentation pathways are common mechanisms used by malignant cells to reduce visibility to CTL. Examples include alterations in the expression of proteasome subunits involved in the processing of endogenous antigens, downregulation, loss, or imbalance in the expression of enzymes such as endoplasmic reticulum aminopeptidase (ERAP)1 and ERAP involved in the loading of peptides into MHC I molecules, the loss of IFN-γ-signaling components resulting in the prevention of MHC I upregulation, and defects in MHC I expression itself, all of which result in changes to the repertoire of antigenic peptides presented to CTL [15,16]. The down-regulation or total loss of cell surface MHC I in particular has been reported in many studies on a wide array of tumours of different origin, revealing tight associations with the level of tumour-infiltrating lymphocytes, disease progression, and overall survival [15,17,18]. Based upon the functional ability of tumours to recover or up-regulate MHC I expression following cytokine treatment, Garrido et al. reported that the alteration of MHC I can be classified into two categories: (1) Irreversible genetic defects, and (2) Reversible epigenetic modifications (Figure 1) [19]. For irreversible genetic defects (“hard” lesions), the most common aetiologies are loss of heterozygosity in chromosomes 6 and 15 harbouring MHC class I genes or the β2-microglobulin (β2m) gene, respectively [19,20,21,22]. In contrast, reversible epigenetic modifications (“soft” lesions) mainly concern deficiencies in the regulation of genes including β2m, MHC I heavy chain and components of the antigen-processing machinery (APM) and can be compensated for through the administration of therapeutic drugs, such as IFN-γ [19,23]. This framework is supported by evidence from both clinical and experimental animal studies [19,24,25]. Hence, it is important to carefully define the molecular mechanisms responsible for a particular alteration in MHC I phenotype prior to designing specific ways to restore in situ tumour MHC I expression.

### 1.2. Inducing MHC I Expression in Cancer

While defects in peptide-MHC I expression in tumours have been examined extensively, strategies to counteract these defects remain scarce and challenging. The majority of MHC I defects in human cancers such as melanoma, gastric, colorectal, and cervical carcinoma, belong to the category of “soft” lesions [26,27]. The repression of gene transcription, such as the lack of MHC I and APM gene expression in tumour cells, can be recovered by histone deacetylation inhibitors and DNA methyltransferase inhibitors, and/or by activation of the interferon (IFN) signalling pathway and stabilization of the NFkB pathway [16,28,29,30,31,32]. Interestingly, combinatory treatment with IFN-γ, or knockout of known inhibitors of NFkB signalling can further exaggerate the effect of histone deacetylation and DNA methyltransferase inhibitors [30,33]. Additionally, combination treatment of these epigenetic modulators is known to enhance MHC I expression in breast cancer patients [33]. These findings indicate the potential of appropriate combination therapies to increase anti-tumour immunity when treating “soft” lesions. 

Concurrently, “hard” MHC class I genetic aberrations exist in 30–40% of human cancers [20,34] correlating with a poor response to cytokine treatments, and acquired resistance to immune checkpoint (anti-PD-1) inhibitors [35]. Alternative approaches have therefore been investigated, such as replacement of defective genes to solve structural alterations in MHC I and APM. The application of viral vectors encoding peptide transporter TAP or β2m genes has been shown to be capable of up-regulating surface expression of MHC I in APM-deficient tumour cells and restoring recognition by antigen-specific CTL [36,37,38,39]. In mouse models of prostate and lung cancer, this gene therapy approach augmented T cell activation and increased T cell numbers within the tumour microenvironment thereby delaying tumour growth and prolonging survival [36,39]. However, this approach is constrained by the local delivery of the gene of interest, which presents particular challenges for tumours situated in internal organs. 

### 1.3. Circumventing MHC I Down-Regulation

#### 1.3.1. TEIPP Antigens

Subsets of CD8^+^ T cells identified by van Hall et al., may be able to recognise ‘hard’ lesions with impaired TAP, tapasin, or proteasome function [40,41,42]. These T cells appear to recognise an alternative peptide repertoire that emerges in residual MHC I as a result of stimulating alternative processing pathways. These alternative peptides are also known as “T cell epitopes associated with impaired peptide processing” (TEIPP). TEIPP antigens have been shown to be exploitable in mouse models of TAP-deficient RMA-S or MCA tumours through vaccination with molecularly identified TEIPP peptides, cellular vaccination with TAP-deficient dendritic cells presenting TEIPP antigens, and adoptive transfer of in vitro expanded TEIPP-specific CTLs [40,43,44]. Vaccine-mediated priming of TEIPP-specific CTLs resulted in efficient homing to the tumour site and subsequently protection against outgrowth of APM-deficient tumours in mice [45]. The potential for clinical translation of these TEIPP antigens has been demonstrated by several studies including the identification of 16 novel human TEIPP peptides by Marijt et al. [46], and the success by Durgeau et al. utilizing a vaccination strategy of pooled peptides (TEIPP included) in a humanized mouse model to control outgrowth of TAP-deficient lung carcinoma cells [47]. These studies shed new light on the clinical exploration of TEIPP antigens and TEIPP-specific CTLs, which could be employed to mitigate the effects of APM-loss in cancer, and show potential for combination with currently available tumour antigens to improve cancer vaccine efficacy.

#### 1.3.2. CAR T Cells

Another way to overcome the problem of MHC I down-regulation on tumour cells is through the use of chimeric antigen receptor T cells (CAR T cells) [48]. These T cells do not rely on peptide-MHC I complex recognition to kill target cells, but instead have been engineered to express antibody receptors that recognize tumour-associated surface proteins. The adoptive transfer of CAR T cells is a promising anti-tumour therapeutic, which has led to high complete response rates in patients with haematological malignancies [49,50,51]. There have been hundreds of clinical trials involving CAR T cells worldwide since the first two US Food and Drug Administration (FDA)-approved CAR T therapies targeting CD19 on B cell malignancies [8,11]. CD19 continues to be the most popular antigen target [10], though patient relapses have been shown in multiple trials due to the emergence of CD19-negative leukaemia cells, a phenomenon known as antigen escape [52,53]. This may explain the low long-term survival rates versus the high initial remission rates in all patients after CD19 CAR T treatments. Further challenges remain for CAR T cell therapy including treatment-associated toxicities [52,54]. Targeting solid tumours with CAR T cells poses additional obstacles including the lack of tumour-specific antigens (ideally antigens must be highly expressed on most tumour cells but absent on normal tissues), the inefficient infiltration of CAR T cells into solid tumour masses, and the immunosuppressive effects of the tumour microenvironment. 

Bispecific CAR T cells that recognize CD19 and an additional antigen, have the potential to offset antigen-loss-associated tumour escape and improve long-term survival rates and are the subject of much interest in the field. In the last 5 years in particular, preclinical animal models have emerged to test the viability of CARs with dual targets, such as CD19/HER2 [55], CD19/CD20 [56], and CD19/CD22 [57], and have shown promise. Grada et al. were amongst the first to construct a single-chain CAR with two distinct antigen recognition domains presented in tandem: CD19 and the human epidermal growth factor receptor 2 (HER2/neu) [55]. This tandem CAR (TanCAR) responded to either CD19 or HER2 and was able to trigger T cell activation, which was confirmed in a xenograft mouse model utilising Daoy.TET.CD19 tumour cells. The Daoy.TET.CD19 cells expressed HER2 constitutively and CD19 only when cultured with doxycycline. The adoptive transfer of TanCAR T cells alone, in the absence of doxycycline, resulted in significantly delayed tumour progression, compared to the group without transferred T cells. The tumour growth was further delayed by induction of CD19 with doxycycline. Hence, simultaneous recognition of two antigens enhanced in vivo anti-tumour activity by these bispecific CAR T cells. However, as HER2 and CD19 are not typically expressed on the same cell, tumour cells can still escape TanCAR detection by eliminating CD19 expression. To generate a bispecific CAR T cell that can control both CD19^+^ B-cell lymphoma and CD19^−^ loss variants with equal efficiency, Zah et al. determined the structural parameters for dual-antigen recognition and created CD19-OR-CD20 CAR, an “OR-gate CAR”, to trigger robust T cell function by either antigen expression [56]. The in vivo functionality of CD19-OR-CD20 CAR has been determined in a NOD SCID Gamma mouse model utilising Raji tumour cells [56]. While CD19 CAR T cells could control the tumour growth of wild-type Raji cells but not mixed populations of wild-type Raji cells and CD19^−^ loss variant cells, OR-gate CAR T cells could efficiently target both. Their work presented the potential of CD19-OR-CD20 CAR as a powerful safeguard against antigen escape in CAR T treatments for wild-type and CD19^−^ loss variant B-cell lymphomas. 

Another bispecific CAR, the CD19/CD22 bivalent CAR, was generated by Qin et al. and tested in a “spike-in” leukaemia relapse model utilising NALM6 cells and CD19^−^/CD22^−^ loss variants edited by CRISPR-Cas9 technology [57]. The inoculum in this B-ALL xenograft model was a mixture of 1% CD19^−^ or CD22^−^ B-ALL with 99% wild-type cells (CD19^+^/CD22^+^), which mimics relapse from a small pre-existent clone as observed in clinical trials. Adoptive transfer of T cells expressing CD19/CD22 bivalent CAR eradicated CD19^+^/CD22^−^ B-ALL xenografts and significantly inhibited the outgrowth of CD19^−^/CD22^+^ B-ALL populations. The potency of this bivalent CAR to reduce the risk of antigen escape was further investigated in the patient-derived xenografts: HMB15 (CD19^+^/CD22^+^) and HMB28 (CD19^−^/CD22^+^). Encouragingly, HMB28 was resistant to CD19 CAR but cleared by the bivalent CAR, while HMB15 was eradicated by either CAR. These clinically relevant models of CD19 CAR resistance have helped to confirm proof-of-concept in vivo activity of bispecific CARs and their potential applicability as adoptive T cell transfer therapies for cancer treatments.

It will be interesting to further utilize the design principles of bispecific CARs highlighted in the studies above to construct novel CARs targeting additional antigen pairs. Examinations of promising alternative targets are underway, such as CD30 in refractory Hodgkin’s lymphoma, BCMA in multiple myeloma, and CD33, CD123, and FLT3 in AML [52]. It remains possible that minor modifications to individual CAR components could further increase the functionality of TanCAR, OR-gate CAR, or bivalent CAR. Several studies have shown the importance of single-chain variable fragment framework regions, costimulatory signals and extracellular spacer length in enabling robust T cell-mediated responses, which provides critical clues for the optimization of bispecific CAR targeting strategies [58,59,60].

A current challenge for CAR T cell therapy is to overcome barriers to their persistence following adoptive transfer. Emerging research indicates that cell-intrinsic mechanisms affect CAR T cell exhaustion, activation-induced cell death, and lymphocyte contraction, suggesting that the production of CAR T cells from less differentiated T cells (i.e., naïve, central memory or stem like T cells) is associated with superior functional properties (reviewed in Ref. [61]). A more comprehensive understanding of CAR signalling, improving CAR trafficking and survival, and scrupulously studying their efficacy for tumour treatments in related preclinical animal models will continually facilitate the development of more effective CAR T therapeutics [48]. 

#### 1.3.3. Gamma-Delta T Cells

An alternative to CAR T cell transfer for the treatment of tumours with down-regulated or defective MHC I expression is the adoptive transfer of γδ T cells. Like conventional αβ T cells, γδ T cells demonstrate cytotoxic activity via the granzyme-perforin axis, release cytokines such as TNF-α, IFN-γ, and IL-17, and can kill a wide array of tumour types [62]. However, unlike conventional αβ T cells, γδ T cells are activated in an MHC-independent manner and are capable of recognising a broad range of stress-induced molecules on cancer cells [63]. The adoptive transfer of γδ T cells into cancer patients is known to be safe and associated with improved overall survival [64], although overall treatment efficacy is highly variable, most likely as a result of the exploration of greatly varied γδ T cell expansion and delivery protocols [62]. While there is much to be learned regarding the optimal methodology for isolating and expanding suitable numbers of γδ T cells for clinical use, the early indications are that γδ T cells may present great potential as an off-the-shelf cancer therapy.

## 2. T cell Recruitment and Regulation

As a co-receptor for the αβ TCR, the CD8 molecule stabilizes TCR-MHC I complexes and provides downstream signalling for T cell activation. Detecting co-expression of CD3 and CD8a molecules can be used to identify the presence of tumour-infiltrating CD8^+^ T lymphocytes [65]. Studies using CD8 immunohistochemical staining have shown that tumour-infiltrating CD8^+^ lymphocytes are favourably associated with patient survival in several cancers (i.e., breast, bladder, colorectal, esophageal, glioma, lung, melanoma, ovary, renal and head and neck cancers) [66,67,68,69,70]. Thus, emerging evidence suggests that the nature of the tumour infiltrate impacts upon the clinical outcome justifying the selection of CD8^+^ T cells as the main therapeutic target in cancer treatments. However, tumour infiltrates can vary widely in composition, and studies have shown that while the infiltration of effector T cells, such as CD8^+^ lymphocytes and M1 macrophages, are associated with a favourable prognosis, the infiltration of Tregs and M2 macrophages are less favourable and may directly stimulate tumour growth [71,72,73,74]. Further research is required to investigate the interactions between intratumoural subsets and the signalling pathways involved in T cell recruitment that favours tumour control.

Large quantities of literature have reported tumour-infiltration of CD8^+^ T cells as a critical factor in anti-tumour immunity. However, the over-activation of CD8^+^ T cells can potentially trigger autoimmune disorders, such as vitiligo when this occurs in the skin. Isolated auto-reactive T cells from vitiligo patients are able to recognise and trigger apoptosis of healthy melanocytes in vitro. Interestingly, this autoimmune tissue destruction has been reported to provide a hospitable niche for the residence of tumour-specific T cells, the necessary components of long-lasting protection against melanoma in peripheral tissue [75]. Tumour-specific T cells are capable of making IFN-γ and proved critical for protection against melanoma rechallenge, based on established mouse models of melanoma-associated vitiligo. In general, to prevent the unwanted destruction of host tissues, CD8^+^ T cell activation is tightly regulated through central and peripheral tolerance mechanisms, but the mechanisms that regulate memory/effector T cell function in the tumour microenvironment are still not well understood. It is of particular interest to elucidate how effector memory T cells are regulated in the context of anti-tumour responses. Furthermore, there is much to be learned about how cancers establish local immune tolerance to evade immune surveillance and how this may be exploited to develop more effective immunotherapies.

### 2.1. T Cell Infiltration and Tumour Regression

T cell-mediated tumour rejection involves several steps: (1) Effector cell activation/sensitization to tumour antigens in lymphoid organs, (2) Migration of antigen-specific T cells along with other effector cells to tumour sites, and (3) Target cell recognition and killing. Tumour rejection can potentially fail whenever any of these steps is suppressed. Above, we discussed the alterations made by tumour cells to interfere with the activation of antigen-specific T cells and strategies that could be utilized to break down corresponding barriers in immunotherapies. Here, the focus is on studies of tumour-infiltrating lymphocytes, and the cytokine/chemokine signalling networks that stimulate their trafficking to the tumour site. 

A key cytokine controlling lymphocyte extravasation is IFN-γ [76]. The presence of IFN-γ is known to correlate with cell growth inhibition, cytotoxicity and anticancer capacity, however, it is still not fully understood how IFN-γ contributes to the promotion of protective tumour immunity. IFN-γ is important for T cell, natural killer (NK) cell, and NKT cell trafficking into tumours through the induction of chemokines such as CXCL9, CXCL10, and CXCL11 [77,78]. Accordingly, in a CSA1M fibrosarcoma model in IFN-γ^−/−^ mice, T cells were shown to fail to migrate to the tumour site [79]. As the receptor to these three IFN-γ-inducible chemokines, CXCR3 expression was greatly enhanced on effector T lymphocytes (CD4^+^ Th cells, CD8^+^ CTLs and memory T cells) as compared to naive precursors [80]. Several studies examining the B16 tumour model in CXCR3^−/−^ mice have indicated the importance of CXCR3 for CTL migration to the tumour site in melanoma [81,82]. Similar models have also highlighted an important role for the expression of CCR5, the receptor for CCL3, CCL4, and CCL5, on both CD4^+^ and CD8^+^ T cells in this process [83]. Clinical data further confirm CCR5 and CXCR3 expression on T cells to correlate with CTL recruitment and a favourable outcome in melanoma in patients [65,84]. Moreover, the strong expression of CXCL9 and CXCL10 within tumour tissues is also associated with high CD8^+^ T cell infiltration in melanoma patients. Thus, CXCR3 cognate chemokines can affect the recruitment of CTLs into the tumour mass and these effector cells can shape tumour immunity and anti-tumour therapy responses by antigen-specific or antigen-independent mechanisms. Recently, it has been proposed that tumour cells can reduce CXCL9/CXCL10 gradients by secreting galectin-3, a lectin that blocks IFN-γ diffusion [85]. Experimental research in several disease models indicates that the lack of these CXCR3 ligands dramatically impairs cell-mediated immunity. On the other hand, however, the CXCR3 axis can also directly impair immunity through the recruitment of Tregs [86]. 

Though lymphocyte infiltration is important for tumour control, blood vessels within tumours are often leaky with intense interstitial pressure effected by the density of the surrounding tumour mass, which may cause heterogeneous permeability and promote irregular blood flow [87]. As a consequence, leukocyte trafficking within the tumour microenvironment may well be more difficult than in healthy tissues. Moreover, endothelial cells in tumour sites can express the endothelin B receptor and its ligand endothelin-1 to trap T cell infiltrates [88]. Angiogenic factors in the tumour microenvironment such as TGF-β, VEGF, FGF, and nitric oxide could also inhibit the upregulation of cell adhesion molecules like ICAM-1, ICAM-2, VCAM-1, and CD34 on endothelial cells, rendering them refractory to immune cell infiltration [89].

### 2.2. T Cell Regulatory Checkpoints

Tumour-specific infiltrates are necessary to eliminate tumours, but the mere presence of an infiltrate itself is not sufficient. The tumour microenvironment is known to contain many immunosuppressive factors suggesting immune regulation plays an active role in tumour progression [90]. The discovery of T cell inhibitory checkpoint molecules provides novel therapeutic targets that can improve tumour-specific T cell immunity in vivo. These checkpoint molecules include cytotoxic T-lymphocyte-associated protein-4 (CTLA-4), programmed cell death protein-1 (PD-1), lymphocyte activation gene 3 (LAG-3), T cell immunoglobulin-3 (TIM-3) and T cell immunoglobulin and ITIM domain (TIGIT), and therapies targeted to these molecules are currently a hotbed of both basic and clinical research [91,92]. However, patients treated with checkpoint inhibitors (as a single agent or in combination) are at high risk of suffering from immune-related adverse events, such as diarrhea, colitis, myocarditis and endocrine disorders, as a result of the non-specific and systemic activation of immune system [91,93]. 

Further insight into the mechanistic differences that exist between major checkpoint pathways could help optimise combination therapies with checkpoint inhibitors. For example, CTLA-4 plays a role in regulation of T cell activation and in Treg-mediated suppression of dendritic cell function, but not in the control of NK-cell activities [91,92]. On the other hand, PD-1 and TIGIT are expressed on activated T cells as well as NK cells, thus regulating T cell activation, NK cell function, and promoting the differentiation of helper T cells into Tregs [94,95]. TIM-3 has a broader expression profile and can be seen on activated T cells, NK cells, dendritic cells and monocytes [96,97]. Rather than directly inhibiting the activity of effector T cells and NK cells, TIM-3 primarily promotes the expansion of immune suppressor cell populations, such as Tregs, macrophages and myeloid-derived suppressor cells [98]. LAG-3 is expressed on both activated T cells and NK cells [99], and is also a marker of type 1 Tregs [100]. By disrupting or knocking out the corresponding genes in mice, the functions of many checkpoint molecules can be defined. For example, the fact that CTLA-4^−/−^ mice have a lethal autoimmune phenotype indicates the critical role of CTLA-4 in promoting self-tolerance [101]. Similarly, mice lacking PD-1 expression also develop an autoimmune phenotype but with delayed onset and less severe conditions, suggesting that PD-1 has an important role in self-tolerance induction following T cell priming, and that it mainly regulates the immune responses in peripheral tissues [101]. 

## 3. T Cell Immunotherapy: Linking Mouse and Human

### 3.1. Why Use Murine Models for Cancer Immunotherapy Research?

Immunotherapy has shown substantial benefit in the treatment of multiple malignancies. Dramatic advances in our understanding of the interactions between the immune system and tumours have revolutionised the development of new immunotherapeutic approaches, aimed at harnessing the immune system to create effective anti-tumour responses. To evaluate the effectiveness of immunotherapeutic strategies in the context of a functional immune system, animal models are required to replicate human diseases. Mice become the first choice as they are relatively low cost, have a short reproductive cycle, can be easily genetically modified with specific characteristics, contain a functional immune system, and exhibit a high rate of tumour growth [102]. However, there are subsets of patients across several cancers who have not shown robust responses to immunotherapies otherwise predicted by mouse models to deliver encouraging results. This section discusses the pros and cons of current cancer models in mice, and their usage as platforms to predict the effectiveness of immunotherapies in the clinic.

Syngeneic tumour models, obtained through transplantation of cultured tumour cell lines into immunocompetent mice, are the earliest and most commonly utilized murine models to investigate anticancer therapeutics [103] (Figure 2). A major benefit of this approach is its short construction time as transplanted tumour cells generally grow within a few weeks [104]. However, this is also its shortcoming as rapid tumour growth rates result in short time intervals within which to analyse anti-tumour immune responses induced by treatments. In addition, tumours lacking genomic heterogeneity are generated in these syngeneic models that use fully progressed and relatively homogeneous tumour cell clones. Tumours are frequently established subcutaneously, as the adjustment of administration routes to more accurately reflect the tumour microenvironment would require technical support for more complicated manipulations and development of special equipment for both transplantation and tumour growth monitoring [105]. Nevertheless, the syngeneic tumour model in immunocompetent mice is still the standard tool to test tumour growth potential in specific settings, due in part to the presence of a fully functional immune system. This makes it available to assess de novo anti-tumour immune responses without the need for adoptive transfer of additional immune populations, which is particularly useful when investigating immuno-oncology agents, such as checkpoint inhibitors. For example, in a syngeneic model utilising an aggressive SCC cell line (KRS-SCC), CD8^+^ tumour-infiltrating lymphocytes were demonstrated to be over-activated and “exhausted” with notable co-expression of PD-1 and LAG-3, but not CTLA-4 [106]. Consistently, dual blockade of PD-1 and LAG-3 significantly inhibited the growth of tumours compared to control mice. Studies such as this highlight the involvement of T cells expressing multiple checkpoint receptors in the progression of SCC, as well as the requirement for specific selection of patients to improve response rates for targeted immunotherapy. Syngeneic mouse models such as this can also be applied to investigate the effectiveness of new targets for CAR T cell therapy where murine cell-derived CAR T cells are used [107]. The caveat, however, is that differences in mouse strain genetics can lead to differences in multiple factors within tumour experiments including toxicity and efficiency, necessitating a cautious approach when interpreting results from a single model [103]. 

Genetically engineered mouse models are produced based on transgenic technologies inducing systemic or tissue-specific expression or deletion of genes such as oncogenes and tumour suppressor genes (Figure 2). In general, de novo tumour formation allows for the organization of a complex tumour microenvironment and the potential for the tumour to undergo processes of immune tolerance, immuno-editing, and/or immunosuppressive processes [108]. Moreover, the interaction between the tumour and the immune system as the tumour develops permits the formation of tumour heterogeneity, which can be further enhanced by affecting genes associated with mismatch repair and genomic stability [109]. However, a challenge can be the need for non-invasive imaging techniques, such as ultrasound or magnetic resonance imaging, to monitor tumour growth/metastasis and assess anti-tumour immune responses [110]. Compared to syngeneic or patient-derived xenograft models, genetically engineered models are infrequently applied when studying the effectiveness of checkpoint inhibitors or CAR T cell therapies. Most often, they are utilized as an advanced platform for understanding and improving potential therapies by revealing aspects of the immune response that could control tumour responsiveness to targeted therapies, chemotherapy, or immunotherapies. 

In carcinogen-induced tumour models, mice bearing tumours induced by carcinogens such as azoxymethane, ultraviolet light, or methylcholanthrene, are directly used to evaluate the efficacy of tumour treatments (Figure 2). These models, including methylcholanthrene-induced and ultraviolet-light-induced fibrosarcomas, were involved in the ground-breaking work by Schreiber and others to identify and investigate immunoediting: a dynamic process that describes how tumours evolve to no longer be recognised and destroyed by the immune system [111,112,113,114]. These models generally require a long time to become established (particularly with respect to induction by ultraviolet light), however they provide a higher level of genomic instability. This complexity also comes with challenges, such as a heavy burden of unknown neoantigens, a lack of shared neoantigens, and in the case of ultraviolet light, immunosuppression. These challenges make it difficult to assess specific immune responses and thus, carcinogen-induced tumour models are rarely used to evaluate the effectiveness of antigen-dependent therapies, such as vaccines or CAR T cells. However, carcinogen-derived tumours are frequently used to develop cancer cell lines that can be studied using syngeneic tumour models. 

Human xenograft models are created to meet the requirement to have models of human tumours interacting with human immune cells (Figure 2). Immunocompromised mice, including athymic nude, severe combined immunodeficiency (SCID), Non-obese diabetic-SCID (NOD-SCID), and recombination-activating gene 2 (Rag2)-knockout animals, are used as hosts for human cell transplantation. These models have particular relevance for the development and assessment of CAR T cell therapies, especially patient-derived xenograft models created by transplanting the tissue or cells from a patient’s tumour into an immunodeficient or humanized mouse. Most studies discussed in Section 1.2 used the human xenograft model, including studies on the generation and application of bispecific CAR T cells, including “TanCAR”, “CD19-OR-CD20 CAR” and “CD19/CD22 bivalent CAR”. The significant strength of this model is that it allows the natural growth of the human tumour, its monitoring, and corresponding treatment evaluations for patients, particularly when the tumours are transplanted orthotopically [115,116]. However, this also becomes its prominent shortcoming as immunodeficient mice must be used to prevent host immune attacks against the xenotransplanted tumour. To reconstitute the immune system for the effective evaluation of immunotherapies like checkpoint blockade, humanized animals are created by engrafting peripheral blood or bone marrow cells into NOD-SCID mice [117]. These strategies permit investigations into the interactions between xenogenic human stroma and tumour environments in the progression and metastasis processes. However, it remains unknown whether the reconstituted immune system behaves in the same manner as it does in the patients. Furthermore, the reconstituted immune system can be ‘hyper-activated’ as a result of exposure to murine tissues, in a similar pattern to graft versus host disease [118]. 

### 3.2. Combining Immunotherapy with Other Cancer Therapies

The main aim of cancer therapies is to destroy the tumour while preserving normal tissues and limiting toxicity as much as possible. Immunotherapy utilizing anti-tumour T cells greatly improves the potential for tumour specificity whilst minimizing off-target effects. The combination of immunotherapy with targeted therapies and chemotherapies may also induce stronger anti-tumour immune responses [120]. Chemotherapeutic drugs, for example, have been reported to induce DNA damage responses, increase stress signals on tumour cells, enhance tumour antigen presentation by dendritic cells, and stimulate the recruitment and anti-tumour activities of T cells and NK-cells [121]. In addition, these therapies have the potential to boost anti-tumour immunity by manipulating or eliminating immunosuppressive cell populations [120,122,123]. In non-small cell lung cancer clinical trials for example, the anti-PD-1 antibodies nivolumab and pembrolizumab, and the anti-PD-L1 antibody atezolizumab, have all proven to be safe when used (individually) in combination with platinum-based chemotherapies, and furthermore, combination arms have been shown to outperform chemotherapy alone, resulting in FDA approval for chemo-immunotherapy combinations as first line therapies [124]. However, combination therapies can also result in increases in immune-related adverse events, including fatigue, dermatologic toxicities, gastrointestinal toxicities and endocrine toxicities, although these adverse events are generally manageable through the administration of corticosteroids [124]. Therefore, while it is increasingly clear that the tumour microenvironment plays a key role in determining the response to therapy [125,126]; there is a need for experimental models that imitate the diverse tumour-immune environments observed in different human cancers when considering new therapeutic combinations.

In spontaneous mouse models of fibrosarcoma, pro-B-cell leukemia, T cell acute lymphoblastic lymphoma, and gastrointestinal stromal tumours, it has been proven that the efficacy of drugs that target tumour oncoproteins depends on the activity of accompanying anti-tumour T cell responses [126,127]. Interestingly, it has been suggested that a T cell response is necessary to completely eliminate tumours treated with targeted therapies due to their capacity not only to destroy tumour cells but also the tumour vasculature [126,127]. CD8^+^ T cells are required to ensure the efficacy of a chemotherapy drug, doxorubicin, which is used to treat carcinogen-induced sarcomas [128]. However, in the treatment of breast cancers, the infiltration of innate immune cells including tumour-associated macrophages results in resistance to chemotherapeutics such as paclitaxel and doxorubicin [129,130]. In these settings, the treatment responses were improved by blocking the recruitment of these immune cell populations to the tumour site. Along with the development of proper animal models for cancer research, new insights into targeted and immune therapies, as well as conventional chemotherapy and radiotherapy, will help to drive the design of combinatorial strategies to improve patient outcomes. 

## 4. Discussion

It is now well established from murine tumour models that T cells can play an important role in tumour control. However, as tumour models are continually refined to more clearly mimic human disease, it is also evident that tumours resist targeting by these T cells. Absence of tumour-specific antigens, regulation of antigen presentation pathways, interference with T cell trafficking and engagement of inhibitory receptors on T cells all contribute to progressive tumour growth. The broad challenge is to understand these immune escape pathways in the context of heterogeneous tumours and find new approaches to overcoming potentially multiple immune escape pathways simultaneously. CAR T cells were first engineered to overcome the single challenge of insufficient peptide/MHC complexes on tumours but now face the need to target multiple tumour antigens together with improved trafficking and survival in the tumour microenvironment. Overcoming multiple tumour-directed challenges requires a multifaceted approach to T cell activation, trafficking and survival in the future. This is increasingly evident with studies on combination immunotherapies and the immune interface with chemo/radiotherapy. Maintaining specificity for the tumour while broadly enhancing multiple T cell attributes will remain a key focus into the future.

## Figures and Tables

**Figure 1 ijms-22-02907-f001:**
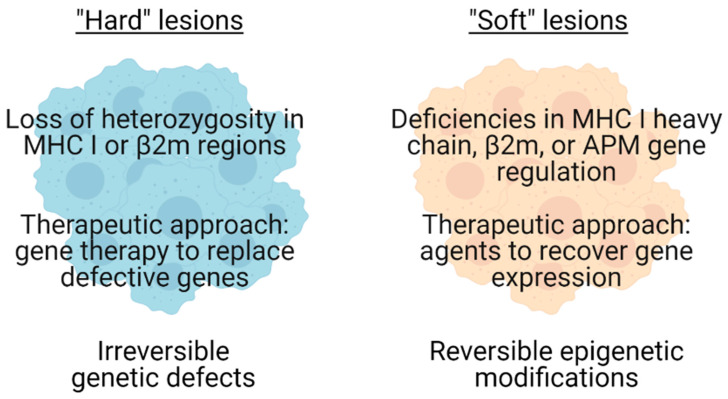
MHC I aberrations in cancer. Tumours can be categorised into “Hard” or “Soft” lesions based on the nature of MHC I defects [19]. Mutations, deletions, or loss of heterozygosity result in irreversible structural defects in the MHC I pathway and can be categorised as “Hard” lesions. Downregulation or inhibition of gene expression result in reversible regulatory defects in the MHC I pathway and can be categorised as “Soft” lesions. Therapeutic approaches differ according to lesion type.

**Figure 2 ijms-22-02907-f002:**
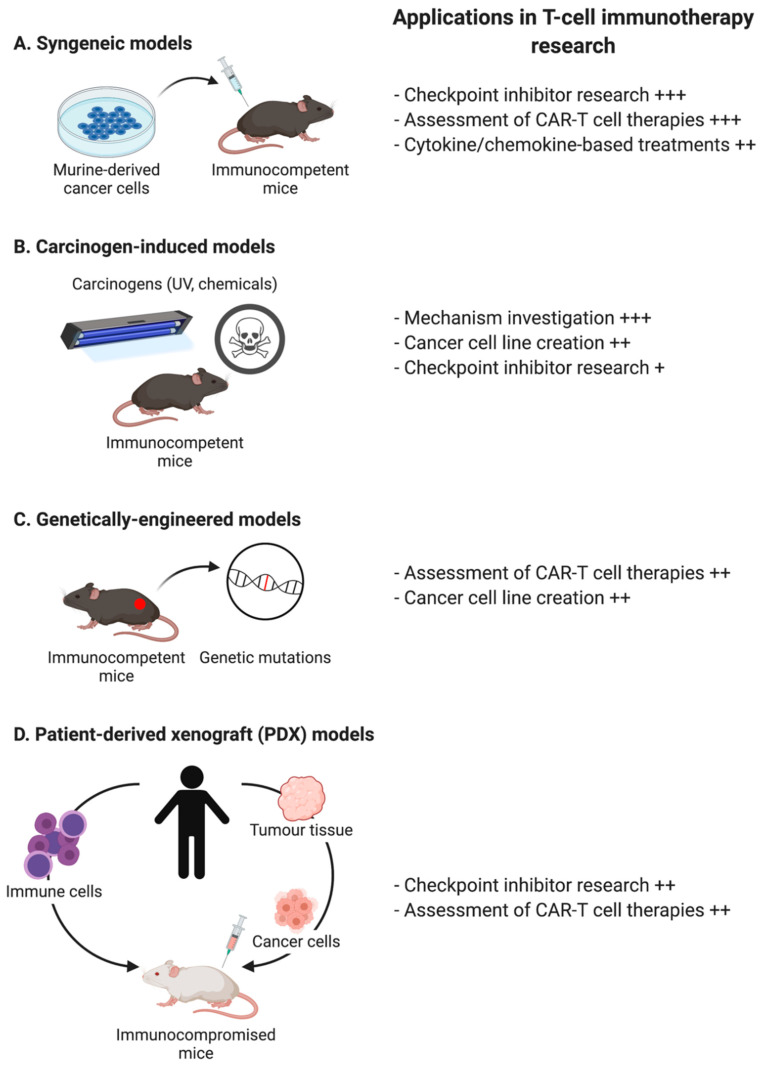
Murine models for evaluation of T cell-directed tumour immunotherapy. (**A**) Syngeneic models utilize murine-derived cancer cell lines grown and expanded in vitro and then injected into immunocompetent mice (commonly either subcutaneously or orthotopically); (**B**) Carcinogen-induced models utilize carcinogens such as ultraviolet light or chemicals to induce mutations resulting in the formation of de novo tumours; (**C**) Genetically engineered models involve transgenic technologies to induce depletion of tumour suppressor genes or expression of oncogenes, which drives autochthonous tumour growth; (**D**) Patient-derived xenograft (PDX) models involve collecting patient tumour material and injecting it into immunocompromised mice. Cancer cells obtained from tumour tissues can be injected alone or in concert with immune reconstitution by autologous immune cells or stem cells. Frequencies are implicated as: +++, frequently; ++, in some cases; +, rarely. Figure inspired by references [103,119].

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
