# Peer review of "Investigating T Cell Immunity in Cancer: Achievements and Prospects"

_ijms, 2021, doi:10.3390/ijms22062907_

Round 1
Reviewer 1 Report
The authors summarize the importance as well as gaps in knowledge of the importance of T cells and T cell-directed therapies in mediating tumor control. In addition to improving endogenous antigen presentation via MHC-I modulation, both adoptive CAR-T therapy as well as immune checkpoint blockade therapy to improve T cell recruitment and function in tumors have resulted in substantial outcomes in both murine models and patients. Despite this, the success has largely been restricted to certain "hot" cancers as well as responses seen only in a subset of patients, highlighting the need to understand the mechanisms underlying T cell checkpoint regulation as well as the harsh tumor microenvironment (TME) where T cells need to persist and mediate tumor regression.
The review is well organized with sufficient background and references provided for the sub topics chosen. However, certain aspects still need to be addressed.
- While the authors address T cell recruitment and checkpoint molecule expression, survival/persistence of T cells in the TME is not addressed. The harsh metabolic changes in the TME (hypoxia and associated acidity, glucose deprivation, abnormal angiogenesis etc) make it difficult for T cells to survive in the TME and undergo either exhaustion or apoptosis. The effect of metabolic alterations in the TME influencing anti-tumor T cell immunity needs to be briefly addressed.
- The role of Tregs, MDSC and TAM in inhibition of anti-tumor T cell function needs to be summarized.
- The section 3.2 on combining immunotherapy with other cancer therapies needs to mention human data on combining immuno- and chemotherapeutic approaches and mention the pros and cons of such combinations.
Author Response
Reviewer 1
- The effect of metabolic alterations in the TME influencing anti-tumor T cell immunity needs to be briefly addressed. (hypoxia and associated acidity, glucose deprivation, abnormal angiogenesis etc)
- Tregs, MDSC and TAM in inhibition of anti-tumor T cell function needs to be summarized
We have now added a new paragraph into section 1 (lines 37-48) to address these two points –
The microenvironment within solid tumours is often hostile to tumour-specific T cells, negatively affecting their function and survival. Metabolic alterations to tumour and stromal cells result in the overexpression of glucose transporters and key glycolytic enzymes, altered growth factor signalling, hypoxic conditions, increases in acidity, and abnormal angiogenesis. Furthermore, an abundance of suppressive immune cells, including regulatory T cells (Tregs), myeloid-derived suppressor cells, and tumour-associated macrophages, infiltrate into the tumour and deliver a host of negative signals to tumour-specific T cells and antigen-presenting cells including dendritic cells. As these features of the tumour microenvironment have been reviewed extensively elsewhere (see Refs [6-11]), we will focus here on an examination of alterations to antigen processing and presentation pathways [12-14], factors influencing T cell infiltration, and strategies designed to improve tumour-targeting by T cells in the sections below.
- The section 3.2 on combining immunotherapy with other cancer therapies needs to mention human data on combining immuno- and chemotherapeutic approaches and mention the pros and cons of such combinations.
We have now added the following new paragraph into section 3.2 as requested (lines 456-469)-
In non-small cell lung cancer clinical trials for example, the anti-PD-1 antibodies nivolumab and pembrolizumab, and the anti-PD-L1 antibody atezolizumab, have all proven to be safe when used (individually) in combination with platinum-based chemo-therapies, and furthermore, combination arms have been shown to outperform chemo-therapy alone, resulting in FDA approval for chemo-immunotherapy combinations as first line therapies [124]. However, combination therapies can also result in increases in immune-related adverse events, including fatigue, dermatologic toxicities, gastrointestinal toxicities and endocrine toxicities, although these adverse events are generally managea-ble through the administration of corticosteroids [124]. HoweverTherefore, while, it is increasingly clear that the tumour microenvironment plays a key role in determining the response to therapy [125, 126];. Again, this highlights thethere is a need for experimental models that imitate the diverse tumour-immune environments observed in different hu-man cancers when considering new therapeutic combinations.
And, we have now altered/updated the final paragraph in section 1.3.2 (lines 199-207) to include mention of less differentiated cells:
A current challenge for CAR T cell therapy is to overcome barriers to their persistence following adoptive transfer. Emerging research indicates that cell-intrinsic mechanisms affect CAR T cell exhaustion, activation-induced cell death, and lymphocyte contraction, suggesting that the production of CAR T cells from less differentiated T cells (i.e. naïve, central memory or stem like T cells) is associated with superior functional properties (re-viewed in Ref [61]). A more comprehensive understanding of CAR signalling, improving CAR trafficking and survival, and scrupulously studying their efficacy for tumour treat-ments in related preclinical animal models will continually facilitate the development of more effective CAR T therapeutics [48].
Reviewer 2 Report
This review discusses some key achievements and challenges of cancer immunotherapy, with a focus on tumour escape mechanisms and strategies that have been investigated to improve anti-tumour T cell function, mainly based on murine models. Comments below are made to improve the structure and overall quality of the review.
Moderate
Generally, the manuscript is lacking figures, which would certainly help to better visualize the content of chapters 1 & 2.
Chapter 1.3.2, two aspects of T cell immunity should be introduced and briefly discussed, which refer to: (i) gamma delta T cells, as an alternative to alpha beta T cells or CAR-T cells, that can recognize cancer antigens other than peptides/MHC complexes. (ii) the selection of the optimal T cell subset for adoptive immunotherapy, as transfer of less differentiated T cells (i.e. naïve, central memory or stem like T cells) is associated with superior functional properties.
Chapter 1.1: Among genes responsible for the processing, and presentation of antigens (i.e TAP1/2), I would also mention the loss of IFN-gamma pathway as another tumour escape mechanism. Chapter 1.2 (lane 80): Would emphasize that MHC-I loss is associated with resistance to immune checkpoint blockade therapy.
Chapter “CAR-T”: The expression CD19- mutant is not appropriate. Would suggest to use either CD19- variant or CD19 loss variant.
Minor
The following abbreviations should be added and defined (lane 456): TIGIT, ERAP, TCR, NSG, NK. The abbreviation NSG should be also defined in the text (lane 151).
Author Response
Reviewer 2
- Add figure into chapter 1 and 2 if possible
We have added a new Figure into Chapter 1 describing the “hard/Soft” categorisation of tumours based upon aberrations in MHC I. The Legend for this figure can be found on lines 77-81.
Figure 1. MHC I aberrations in cancer. Tumours can be categorised into “Hard” or “Soft” lesions based on the nature of MHC I defects [19]. Mutations, deletions, or loss of heterozygosity result in irreversible structural defects in the MHC I pathway and can be categorised as “Hard” lesions. Downregulation of inhibition of gene expression result in reversible regulatory defects in the MHC I pathway and can be categorised as “Soft” lesions. Therapeutic approaches differ ac-cording to lesion type.
- Chapter 1.3.2, two aspects of T cell immunity should be introduced and briefly discussed, which refer to: (i) gamma delta T cells, as an alternative to alpha beta T cells or CAR-T cells, that can recognize cancer antigens other than peptides/MHC complexes. (ii) the selection of the optimal T cell subset for adoptive immunotherapy, as transfer of less differentiated T cells (i.e. naïve, central memory or stem like T cells) is associated with superior functional properties.
We have now included a new section in which we discuss gamma-delta T cells as requested (lines 208-221):
1.3.3. Gamma-delta T cells
An alternative to CAR T cell transfer for the treatment of tumours with down-regulated or defective MHC I expression is the adoptive transfer of γδ T cells. Like conventional αβ T cells, γδ T cells demonstrate cytotoxic activity via the granzyme-perforin axis, release cytokines such as TNF-α, IFN-γ, and IL-17, and can kill a wide array of tumour types [62]. However, unlike conventional αβ T cells, γδ T cells are activated in an MHC-independent manner and are capable of recognising a broad range of stress-induced molecules on cancer cells [63]. The adoptive transfer of γδ T cells into cancer patients is known to be safe and associated with improved overall survival [64], although overall treatment efficacy is highly variable, most likely as a result of the explo-ration of greatly varied γδ T cell expansion and delivery protocols [62]. While there is much to be learned regarding the optimal methodology for isolating and expanding suit-able numbers of γδ T cells for clinical use, the early indications are that γδ T cells may present great potential as an off-the-shelf cancer therapy.
And, we have now altered/updated the final paragraph in section 1.3.2 (lines 199-207) to include mention of less differentiated cells:
A current challenge for CAR T cell therapy is to overcome barriers to their persistence following adoptive transfer. Emerging research indicates that cell-intrinsic mechanisms affect CAR T cell exhaustion, activation-induced cell death, and lymphocyte contraction, suggesting that the production of CAR T cells from less differentiated T cells (i.e. naïve, central memory or stem like T cells) is associated with superior functional properties (re-viewed in Ref [61]). A more comprehensive understanding of CAR signalling, improving CAR trafficking and survival, and scrupulously studying their efficacy for tumour treat-ments in related preclinical animal models will continually facilitate the development of more effective CAR T therapeutics [48].
- Chapter 1.1: Among genes responsible for the processing, and presentation of antigens (i.e TAP1/2), I would also mention the loss of IFN-gamma pathway as another tumour escape mechanism. Chapter 1.2 (lane 80): Would emphasize that MHC-I loss is associated with resistance to immune checkpoint blockade therapy.
Re: IFN-gamma pathway: we have now amended the relevant sentence (lines 52-58) to read: “Examples include alterations in the expression of proteasome subunits involved in the processing of endogenous antigens, downregulation, loss, or imbalance in the expression of enzymes such as endoplasmic reticulum aminopeptidase (ERAP)1 and ERAP involved in the loading of peptides into MHC I molecules, the loss of IFN-γ-signaling components resulting in the prevention of MHC I upregulation, and defects in MHC I expression itself, all of which result in changes to the repertoire of antigenic peptides presented to CTL [15, 16].”
Re: MHC I loss and resistance to immune checkpoint blockade therapy: We have now added the following text and reference into the sentence in line 80 (now lines 96-98): “Concurrently, “hard” MHC class I genetic aberrations exist in 30-40% of human cancers [20, 34] correlating with a poor response to cytokine treatments, and acquired resistance to immune checkpoint (anti-PD-1) inhibitors [35].
- Chapter “CAR-T”: The expression CD19-mutant is not appropriate. Would suggest to use either CD19- variant or CD19 loss variant.
We have changed “mutant” to “loss variant” in lines 148, 153, 156, and 159 (now lines 166, 171, 174, and 177) in line with the fact that the authors created targeted CD19-negative (and CD22-negative) cells using CRISPR/Cas9-mediated gene editing
- The following abbreviations should be added and defined (lane 456): TIGIT, ERAP, TCR, NSG, NK. The abbreviation NSG should be also defined in the text (lane 151).
We have made the requested additions/definitions to the Abbreviations section at the end of the document for TIGIT, ERAP, TCR, and NK (line 456, now line 503). Since the abbreviation NSG is only used once in the text, we have elected to spell out the abbreviation instead, see “NOD SCID Gamma” in line 151 (now line 169)